# Soup Kitchen: Cooking Exotic Model Soups across Labels, Losses, Tasks, and Data

## Abstract

Model soups take a model (the stock), fine-tune it into multiple models (the ingredients), and then mix their parameters back into one model (the soup) to improve predictions. That this is possible is surprising and surprisingly effective, so how to fine-tune and mix deserves closer examination to keep converting more train-time computation into more improvement. In this work we identify and analyze novel fine-tunings and mixtures to produce new and exotic soups for visual recognition that nevertheless work. While all known soups require supervised learning, and optimize the same loss on labeled data, our recipes for self-*soup*ervision generalize soups to self-supervised learning and other label-free variations. We show for the first time that 1. ingredients can be fine-tuned without labels by self-supervision and differ by self-supervision hyperparameters (e.g. masking rate), 2. soups can be mixed across losses whether supervised or self-supervised (e.g. MAE and MoCoV3), 3. soups can be mixed across 10+ tasks and partitions of the training data, and 4. ingredients can be fine-tuned by self-supervision on the test data to adapt the soup and improve predictions. Our new soups achieve $1-3\%$ robustness gains on ImageNet variants and up to $10\%$ transfer gains on VTAB with remarkable consistency across our exotic ingredients from self-supervising, partitioning, and adapting to the test data.

## 1 Introduction: More Soups with Less Supervision

Model soups split a single model (the stock), into multiple models (the ingredients) by fine-tuning, then merge them back into one model (the soup) by mixing parameters to improve prediction accuracy (Wortsman et al., 2022). Each fine-tuning varies in its configuration (e.g. optimization hyperparameters) and each mixing can be a simple average or more sophisticated linear combination. In this way, soups convert more training time into more accuracy without more inference time: the soup model requires only as much computation as the original model.

Model soups are surprisingly possible, in that mixing model parameters is absolutely not guaranteed to result in a better model (or even an equally good model!), and surprisingly productive, in that improvements have been shown across multiple domains (vision (Jain et al., 2023), language (Ablin et al., 2025; Jang et al., 2023; Chronopoulou et al., 2023), text-to-image (Biggs et al., 2024), federated learning (Chen et al., 2024)), and benchmarks (image classification (Wortsman et al., 2022), domain generalization (Ramé et al., 2023; Ramé et al., 2022)). However the existing recipes are few, and all known soups are restricted to fine-tuning by supervised learning, so there is still much to understand about (1) when souping is possible and does not hurt results, and (2) how to make souping productive to help results. More specifically, there is potential for more and better soups without the constraints of supervised learning: limited scalability due to the expense of labeling (on large datasets collected from the internet), limited transfer due to differences in the upstream/source task and downstream/target task (across pre-trained models and practical tasks), and limited feasibility in the few-shot regime of sparse labels (in scientific domains like microscopy or remote sensing).

We identify new and different dimensions for souping by self-soupervision: the fine-tuning and mixing of soups with and *without* supervised learning. In particular, we harness self-supervised learning, partitioning data, and adding parameter noise to expand the palette of model soups. By doing so, we fine-tune and mix more models with fewer labels, and for the first time we fine-tune and mix models with no labels at all. Self-soupervision therefore enables more soups than before.

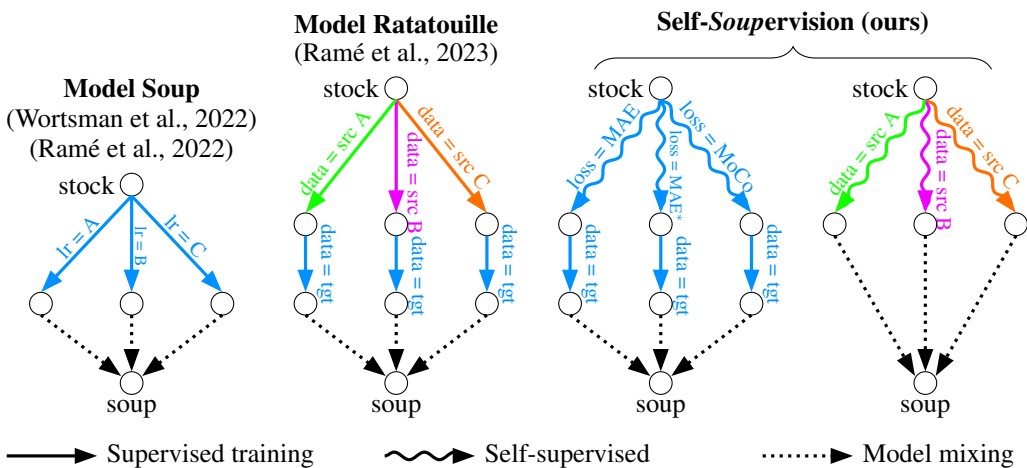

Figure 1: **Soups & Supervision**. Soups fine-tune then mix models to improve predictions. The original soups fine-tune across hyperparameters for the task, while model ratatouilles first inter-train on different data then fine-tune for the task. Both are supervised and cannot harness unlabeled data. Our self-*soup*ervision inter-trains across different losses and data to make more and better soups with and without labels. Our self-soups fine-tune then mix into a supervised model for the task (left) or inter-train then mix into a self-supervised model for transfer (right).

Self-soupervision makes soups strictly more general, in fine-tuning and mixing supervised and self-supervised models, to add flavor while preserving the original recipes. In particular, the union of supervised and self-supervised ingredients can be mixed into a *soup*erset of more soups. Our experiments on accuracy with ImageNet, robustness on ImageNet variants, and transfer across VTAB show more diverse soups is possible, and that this diversity is productive with gains due to our ingredients from self-supervision, partitioning data, and adding parameter noise.

## 2 BACKGROUND: SUPERVISED SOUPS AND SELF-SUPERVISED LEARNING

### 2.1 SUPERVISED MODEL SOUPS

**Initializing cooking with a stock.** Soups require that the mixed models (= ingredients) share the same initial parameters for optimization (the stock). Existing work chooses only supervised stocks Wortsman et al. (2022); Ramé et al. (2022); Ramé et al. (2023). We choose self-supervised stocks.

**Cooking the stock to add ingredients by fine-tuning.** Multiple fine-tunings of the stock on a target/downstream task creates multiple ingredients for mixing a soup. Multiple *different* fine-tunings are key to provide different ingredients: soups rely on differences in the models for their gains. To vary their ingredients, model soups (Wortsman et al., 2022) varied optimization parameters, such as the learning rate, data augmentation, and optimizer.

**Mixing ingredients.** Once the ingredients are prepared, they can be mixed by a simple uniform average, a greedy search, or parallelized methods like "seasoning" which optimizes the mixture coefficients by exhaustive search (Croce et al., 2023) or random sampling.

**Boosting ingredient diversity via inter-training.** Model Ratatouille (Ramé et al., 2023) fine-tunes in two stages, optimizing ingredients for longer, and increasing their diversity for domain generalization. Specifically, Ratatouille first initializes with a stock and "inter-trains" on up to 5 auxiliary labeled datasets independently. Then, these models are fine-tuned on the target task and mixed. Ratatouille gains 0.5% when tested out of distribution. Although this gain is modest, like souping, Ratatouille produces a single model thus does not increase inference/deployment costs; furthermore, souping models trained on different datasets was not known to be possible before Ratatouille.

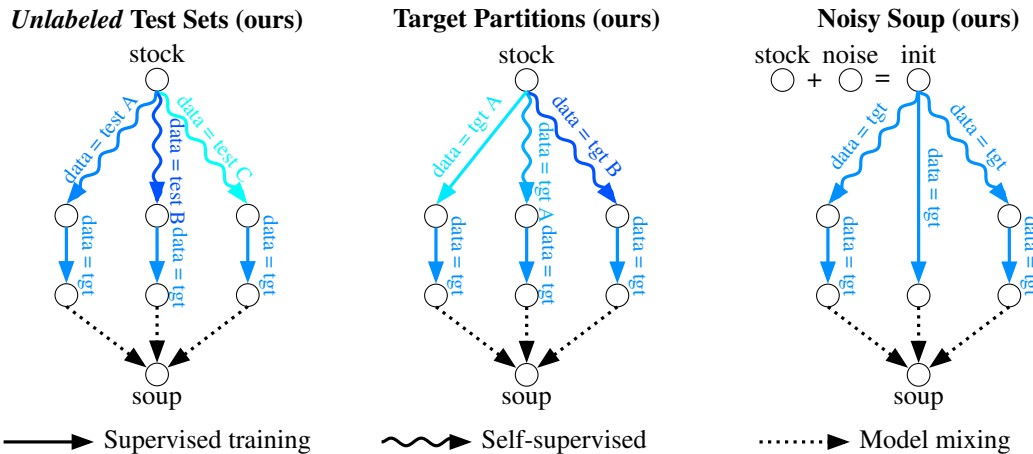

Figure 2: We identify three new soup recipes. (Left) We inter-train using SSL on different test sets, then fine-tune on the target training set, and mix; our self-*soup*ervision makes this possible. (Center) We partition the target training set, inter-train on them independently (with or without labels), fine-tune on the target, and mix. (Right) We add parameter noise to the stock, thus cook soups *without a shared initialization* to create ingredient diversity in a simple manner.

## 2.2 LINEAR MODE CONNECTIVITY

Not all models can be mixed. When mixing works, for ingredients that share a stock, the condition of linear mode connectivity holds. Linear mode connectivity (LMC) holds for two models if the accuracy of the interpolated model weights is greater than or equal to the interpolated accuracy of the models. Formally, for all $\lambda \in [0, 1]$,

$$\mathbf{acc}\left((1 - \lambda) \cdot \theta_A + \lambda \cdot \theta_B\right) \geq \\ (1 - \lambda) \cdot \mathbf{acc}(\theta_A) + \lambda \cdot \mathbf{acc}(\theta_B) \tag{1}$$

where $\lambda$ is the interpolation weight, $\theta_A, \theta_B$ are the model weights, and $\mathbf{acc}$ is the accuracy.

Although model soups can provide gains, the known cases in which LMC holds are few. All methods initialize from a shared stock, but differ in how they optimize to make add ingredients:

- Fine-tuning with *different stochasticity* (e.g., order, augmentation) (Frankle et al., 2020)
- Fine-tuning with *different hyper-parameters* (e.g., learning rate) (Wortsman et al., 2022)
- Inter-training on *different supervised datasets*, then fine-tuning on a target task (Ramé et al., 2023)
- Fine-tuning large language models with *different rewards* for RLHF (Ramé et al., 2023)
- Fine-tuning on a target task with *different adversarial attacks* (Croce et al., 2023)

## 2.3 SELF-SUPERVISED LEARNING

Self-supervised learning (SSL) is a powerful framework because it enables learning from raw/unlabeled data itself; this allows for scaling to gigantic datasets and enables deep learning for niche applications with few annotations (Balestriero et al., 2023). The two most common types of SSL algorithms are reconstructive and contrastive learning. Reconstructive methods hide or noise parts of samples and pre-train models to predict the originals. Contrastive methods pre-train models to produce similar embeddings derived from positive pairs of inputs (e.g., different augmentations of the same sample) and dissimilar embeddings for negatives (e.g., different samples). These are two fundamentally different algorithms that learn different representations (Park et al., 2023). MAE (He et al., 2022) and MoCoV3 (Chen et al., 2021) are simple and popular instantiations of reconstructive and contrastive SSL, respectively. We thus choose these algorithms to experiment with model soups

trained with different losses. We choose MAE over MoCoV3 for all other experiments because it is more popular, more robust to hyperparameters, and does not require large batches.

**Algorithmic Parameters.** Each SSL algorithm has its own configuration. For example, the configuration for MAE defines a masking ratio for how much of the input to mask and reconstruct. The configuration for MoCoV3 (and all contrastive methods) carefully sets the batch size, as negatives come from the batch, so larger batches are harder. These algorithmic design choices result in different learned representations, which enables more diverse ingredients for our self-soupervision.

## 3    METHOD: LABEL-FREE SOUP INGREDIENTS AND INSTANT SEASONING

We cook exotic soups of new ingredients by learning from self-supervised losses, partitioning data, and perturbing parameters with noise. We mix soups from our ingredients without labels (the self-supervised setting) or from our ingredients with different labels (the transfer setting) by searching for mixtures to yield a model compatible with nonparametric prediction by nearest neighbors. We first explain our techniques for making the ingredients and our technique for mixing, in this section on the general methods, and then examine the results for each technique in isolation, in combination, and in addition to existing soup ingredients (from supervised fine-tuning and inter-training), in the following section on the specific experiments. Our techniques serve to enrich the set of possible model soups: by identifying new options for ingredients, we enable new recipes, and increase the number and diversity of settings in which soups are possible and productive.

**Freedom from Labels through Self-*Soup*ervision.** We introduce self-*soup*ervision, which creates ingredients (parameters to mix) by training from stocks (parameters to initialize). Our framework is broad; any use of SSL to prepare ingredients qualifies; thus, there are endless possible instantiations of self-*soup*ervision. For example the choice of stock, SSL algorithm and algorithmic parameters, training data, training length/schedule, optimization hyperparameters, additional training stages (e.g., fine-tuning), etc. Centrally, self-*soup*ervision allows for cooking soups on more data and from new sources—e.g., that are closer to the target distribution—and using different losses—e.g., that are more aligned with the target task. Formally, we define self-*soup*ervision as:

$$\theta = \frac{1}{N \cdot M} \sum_{j=0}^{N-1} \sum_{i=0}^{M-1} \text{Train}(\ \overbrace{\text{Train}(\theta^{\text{pt}}, x_i)}^{y_i \text{ not required}}\ , x_j, y_j) \tag{2}$$

where $\theta$ are soup parameters, $\theta^{\text{pt}}$ are pre-trained parameters, $N$ is the number of fine-tunings per inter-training, $M$ is the number of inter-trainings, $x_i/x_j$ are the inputs of the $i^{th}/j^{th}$ dataset, and $y_j$ are the labels of the $j^{th}$ dataset. **Blue** components alone make vanilla soups. We introduce the **orange** components—which do not require labels—and which we can use alone to make soups without fine-tuning to a task, we show this via transfer in Section 4.5.

**Self-*Soup*ervision on *Unlabeled* Test Data or Training-set Partitions.** Since our self-*soup*ing does not need labels, this enables cooking soups that have seen test-set inputs for improved accuracy on test data. In other words, in equation 2 our $M$ datasets can be test sets, since we do not require their labels $y_i$. Specifically, we initialize from a stock and run several inter-trainings using SSL on the test set—without test labels. We then fine-tune each model on the training set using supervised learning—and we mix these final ingredients for our soup. The core idea is that having cooked on test data, the soup can better generalize from making predictions on training data to test data. Self-*soup*ing on test data may seem heretical, however, other established methods also optimize over unlabeled test data (e.g., test-time adaptation and domain adaptation). We can also cook soups on training-set partitions to create diverse ingredients without auxiliary datasets, which Ratatouille requires (Ramé et al., 2023). Our partitioning method splits the downstream/target training data and inter-trains on them separately before fine-tuning on the full training set.

**Noisy Soups.** Another way to create diversity among ingredients is to add parameter noise. We can inject noise at many stages of cooking, for example at initialization ($\epsilon$ in eq. 3) —which results in soups initialized from different parameters, *a first for soups*. We can also inject noise between inter-training and fine-tuning ($\epsilon$ in eq. 3), or directly before mixing ingredients ($\epsilon$ in eq. 3). Our noisy-soup method is thus a novel and simple kitchen aid.

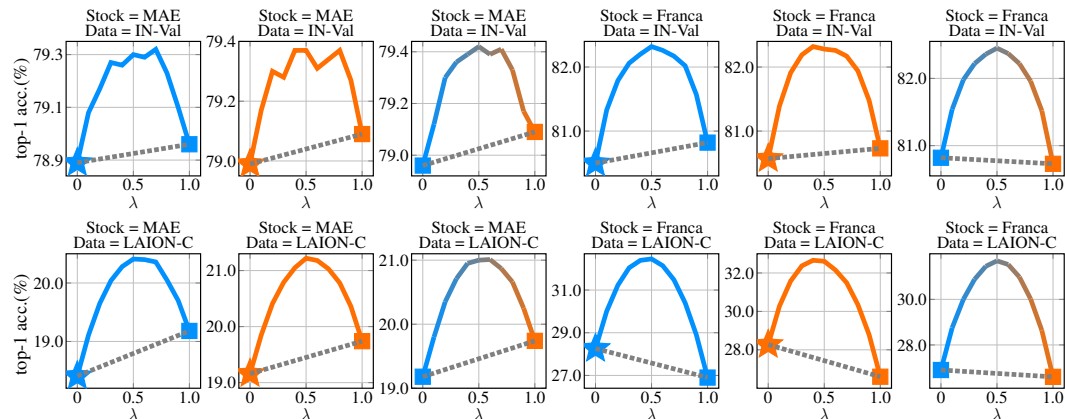

Figure 3: Linearly combining models independently trained using *SSL* (= ingredients) improves accuracy over ingredients alone. We first train models using SSL, initializing from MAE or Franca's ViT-B, then fine-tune on ImageNet-1K to make soup ingredients. We mix ingredients: $\lambda \in 0{\to}1$. It was previously unknown whether effective soups could be made with ingredients trained using SSL.

$$\theta = \frac{1}{N \cdot M} \sum_{j=0}^{N-1} \sum_{i=0}^{M-1} \epsilon + \text{Train}(\epsilon + \text{Train}(\epsilon + \theta^{\text{pt}}, x_i), x_j, y_j) \tag{3}$$

**Mixing by Instant Seasoning.** The original and simplest way to mix a soup is to take the average of the ingredients. This uniform mixture may improve predictions, but may not be the best mixture for a given task and target dataset. Seasoning Croce et al. (2023) searches for a better mixture over a grid of options by mixing each model, making predictions on a few-shot labeled dataset, and picking the best. While effective, this only applies to supervised soups: seasoning makes predictions by mixing the classifiers. We mix ingredients without labels (the self-supervised setting) or with different labels (the transfer setting) that lack classifiers for the target task and so cannot directly compute predictions. We instead mix these ingredient into a model for representation—rather than prediction—then compute its representation on training and testing data for inference by nearest neighbors. To do so we have to divide the few-shot labeled dataset for seasoning into disjoint sets for retrieval and for evaluation. Our variant of seasoning mixes without classifier training for the "instant" seasoning of soups without labels and across different tasks.

## 4 EXPERIMENTS

We experiment using the gold standard of visual recognition benchmarks to confirm that our novel recipes can make effective model soups. These benchmarks include common ImageNet test sets: ImageNet-Val (Russakovsky et al., 2015), -ReaL (reassessed labels (Beyer et al., 2020)), -v2 (Val reproduction (Recht et al., 2019)), -A (adversarial (Hendrycks et al., 2021b)), -HR (high-res (Fuller et al., 2024)), -R (rendition (Hendrycks et al., 2021a)), -C (corruption (Hendrycks & Dietterich, 2019)), and LAION-C (corruption (Li et al., 2025)). They also include datasets from VTAB (Visual Task Adaptation Benchmark (Zhai et al., 2020)), which contains three types of datasets (natural, specialized, and structured) to test transfer.

### 4.1 TESTING LINEAR MODE CONNECTIVITY BETWEEN SELF-SUPERVISED INGREDIENTS

Initialized from a stock, we run four independent inter-trainings, each for 20 epochs on ImageNet (without labels). Two inter-trainings use MAE (He et al., 2022); one with a 25% masking rate and 1 decoder layer, the other with a 90% masking rate and 8 decoder layers. The other two inter-trainings use MoCoV3 (Chen et al., 2021); one with a 512 batch size and a 0.2 temperature, the other with a 2048 batch size and a 1.0 temperature. We then fine-tune each inter-training supervised on ImageNet, which results in four ingredients. We chose these configurations to investigate if LMC

holds with ingredients created using different SSL parameters and different SSL algorithms. We experiment with a ViT-B MAE stock and a ViT-B Franca (Venkataramanan et al., 2025) stock.

**LMC can hold within and across SSL algorithms.** Figure 3 plots the accuracy of two-ingredient soups in distribution (ImageNet-Val) and out of distribution (LAION-C); in all plots, the soup outperforms the interpolated accuracies. This finding is robust to the choice of stock (MAE and Franca). For both stocks, souping gains are greater out of distribution than in distribution.

## 4.2 MIXING ACROSS TEST DATA DISTRIBUTIONS VIA SSL INTER-TRAINING

Initialized from the MAE stock, we inter-train models resulting from the Cartesian product: learning rate {1e-5, 3e-5}, steps {10K, 100K}, data {IN-Val, IN-V2, IN-HR, IN-A, IN-R, IN-C, LAION-C}. This results in 28 models inter-trained on separate test-data distributions *without labels*. We then fine-tune each model on ImageNet for 10 epochs.

**Comparisons.** To add context to the performance of our self-soups, we make vanilla model soups by fine-tuning on ImageNet for 10 epochs and vary optimization parameters between runs: learning rate {5e-5, 1e-4, 2e-4}, weight decay {0.01, 0.05}, stochastic depth {0.0, 0.1}. Note that we do not vary weight decay and stochastic depth for our self-soups, but we could to increase performance further, as self-souping can strictly create more ingredients than supervised soups.

Table 1: **Self-*soup*ervision on test sets without labels.** Since our self-*soup*ervision does not require labels, this allows us to inter-train on test data, then fine-tune on the ImageNet training set. We show that self-*soup*ing across test-set distribution shifts is possible and can be productive. Preparing soups on test data results in large gains, e.g., inter-training on LAION-C increases test accuracy on LAION-C by 9.3% over inter-training on IN-Val. To highlight this effect, we color cells which report on a test set used for inter-training.

| Method | # ing. | IN-Val | IN-ReaL | IN-V2 | IN-HR | IN-A | IN-R | IN-C | LAION-C |
|---|---|---|---|---|---|---|---|---|---|
| Vanilla soup (LR) | 3 | **79.2** | **85.4** | 67.6 | 87.2 | 6.2 | 29.5 | 31.4 | 27.6 |
| Vanilla soup (LR×WD) | 6 | 79.0 | **85.4** | 67.4 | 87.0 | 6.7 | 29.9 | 32.1 | 28.1 |
| Vanilla soup (LR×WD×SD) | 12 | 78.0 | 84.8 | 66.8 | 86.3 | 6.5 | 29.2 | 33.9 | 30.0 |
| self-soup on IN-Val | 4 | 79.1 | 85.3 | 67.7 | 87.0 | 6.3 | 28.7 | 31.7 | 26.6 |
| self-soup on IN-V2 | 4 | 78.9 | 85.2 | 67.7 | 87.0 | 6.3 | 28.8 | 31.6 | 26.5 |
| self-soup on IN-HR | 4 | 79.0 | 85.3 | 67.5 | 87.3 | 6.2 | 28.8 | 31.7 | 26.6 |
| self-soup on IN-A | 4 | 79.1 | 85.3 | 67.5 | 87.1 | 6.7 | 28.9 | 31.7 | 26.6 |
| self-soup on IN-R | 4 | **79.2** | **85.4** | 67.9 | 87.0 | 6.3 | **30.8** | 31.8 | 26.2 |
| self-soup on IN-C | 4 | **79.2** | 85.3 | 67.8 | 87.4 | 6.8 | 28.9 | **35.2** | 27.3 |
| self-soup on LAION-C | 4 | 79.1 | 85.2 | 67.8 | 87.3 | 6.3 | 28.9 | 32.1 | **35.8** |
| All self-soup | 28 | 79.1 | 85.3 | 67.9 | 87.0 | 6.7 | 29.6 | 33.2 | 28.9 |
| All self-soup + vanilla soup | 40 | 78.9 | 85.2 | 67.5 | 86.9 | 6.7 | 29.5 | 33.8 | 29.5 |

**SSL inter-training on test-set inputs helps and self-souping further helps.** Table 1 shows that self-soups inter-trained on separate test distributions is possible. It also shows that self-soups inter-trained on a given test set tend to improve accuracy on the test set. This effect is most clear on the out-of-distribution test sets IN-R, IN-C, and LAION-C. This can be explained by inter-training on test data alone, self-souping alone, and their interaction. For example, self-souping on a test set alone, e.g., IN-Val, gains 3.4% on LAION-C over the stock fine-tuned with the same hyperparameters. Inter-training alone on LAION-C gains 7.7% on LAION-C, on average. The combination of these, that self-soups on the target test set gains 12.6% on LAION-C, which is greater than the gains of the individual strategies added—thus self-souping and test-set inter-training can gain synergistically. Please see the appendix Table 3 for these results and more.

**Ingredient diversity improves robustness.** Figure 4 shows that prediction disagreement (between the two ingredients that are mixed) is correlated with soup gains (soup accuracy minus interpolated accuracy). Interestingly, for a given amount of ingredient diversity, measured by prediction disagreement, soups that saw more test data tend to gain more from souping.

## 4.3 MIXING ACROSS SUPERVISION AND TRAINING-DATA PARTITIONS

Initialized from the MAE stock, we inter-train models on ImageNet training-set *partitions*. We construct four data partitions by grouping classes according to their indices [0, 249], [250, 499], [500, 749], and [750, 999], and include all images belonging to these respective class ranges. For

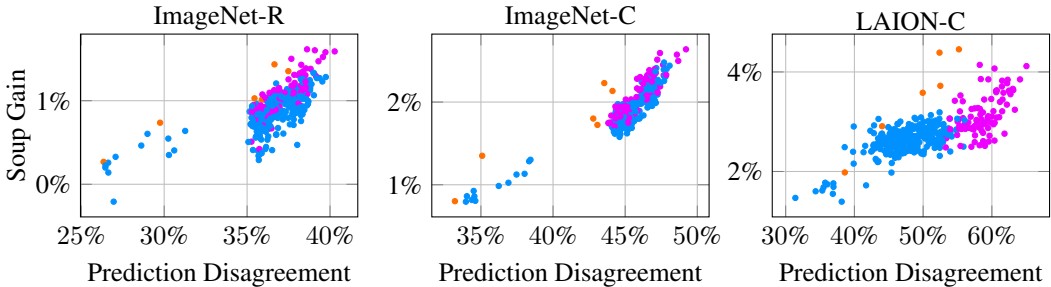

Figure 4: **Ingredient diversity helps and most when inter-trained on test.** Each point is a two-ingredient soup. When both ingredients are inter-trained on the target test data it is **orange**; when one is inter-trained on the target it is **pink**; when neither ingredient sees test data it is **blue**. For a given amount of ingredient disagreement, inter-training on test data tends to increase the gains due to mixing them. Test-set inter-training and self-*soup*ing can thus be synergistic. Soup gain is the soup accuracy minus the average accuracy of the ingredients mixed to create it.

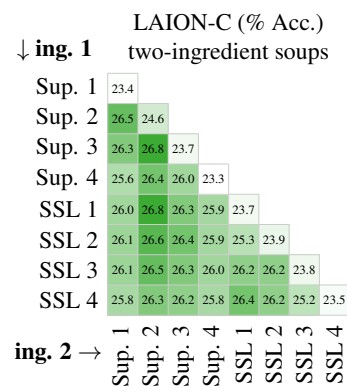

Table 2: **Ingredients prepared on training-set partitions.** Our new recipe that partitions the target training set for inter-training ingredients is both possible and helpful in improving robustness. We show that partitioning works with ingredients inter-trained with supervision, our self-*soup*ervision, and their union.

| Method | # ing. | IN-Val | IN-ReaL | IN-V2 | IN-HR | IN-A | IN-R | IN-C | LAION-C |
|---|---|---|---|---|---|---|---|---|---|
| Vanilla (LR) | 3 | 79.2 | 85.4 | 67.6 | 87.2 | 6.2 | 29.5 | 31.4 | 27.6 |
| Supervised (LR×partition) | 12 | 78.9 | 85.4 | 67.5 | 87.1 | 6.6 | 29.9 | 32.4 | 28.0 |
| Self-*Soup* (LR×partition) | 12 | 78.9 | 85.3 | 67.5 | 87.1 | 6.6 | 29.6 | 32.3 | 28.3 |
| All (LR×partition×loss) | 24 | 78.8 | 85.3 | 67.3 | 86.9 | 6.5 | 29.7 | 32.4 | 28.2 |

Figure 5: (Left) Souping across partitions of the target training set consistently improves robustness over individual ingredients, whether both ingredients use supervised inter-training, both use self-supervised inter-training, or one uses each approach. (Right) Souping over all partitions (28.3%) further improves robustness over two-ingredient partition soups (the best achieves 26.8%).

each partition, we inter-train two models; one supervised by labels and the other self-supervised by MAE. We evaluate on the ImageNet test sets with two-ingredient soups for all 36 combinations, as well as a supervised-only soup, a self-supervised-only soup, and a soup made from of all 8 ingredients.

**Soups can use ingredients from different training-set partitions *and* across supervision types.** Figure 5 shows that two-ingredient soups—where the ingredients differ w.r.t. inter-training data partition or supervision or both—consistently improves accuracy. Furthermore, we can soup all 8 ingredients for further gains, especially under corruption shifts (IN-C and LAION-C).

## 4.4 Noisy Soups: Ingredient Diversity through Adding Parameter Noise

We first compute the mean and std of the MAE-stock parameters; its mean is $\approx 0$ and std is $\approx 0.067$. We inter-train using MAE for 100K steps on the ImageNet training set—crucially, before inter-training we add noise to the initialization by sampling from a gaussian: $\theta_{init} = \theta_{MAE} + \epsilon$, where $\epsilon \sim \mathcal{N}(0, (\mathcal{M} \cdot 0.067)^2)$ and $\mathcal{M}$ is the std multiplier. We sweep $\mathcal{M} \in \{0.01, 0.03, 0.1, 0.3, 1.0\}$ to study the effect of noise amount on souping. To confirm our noisy soups are possible and helpful without self-*soup*ervision, we also run this sweep without inter-training; these experiments add

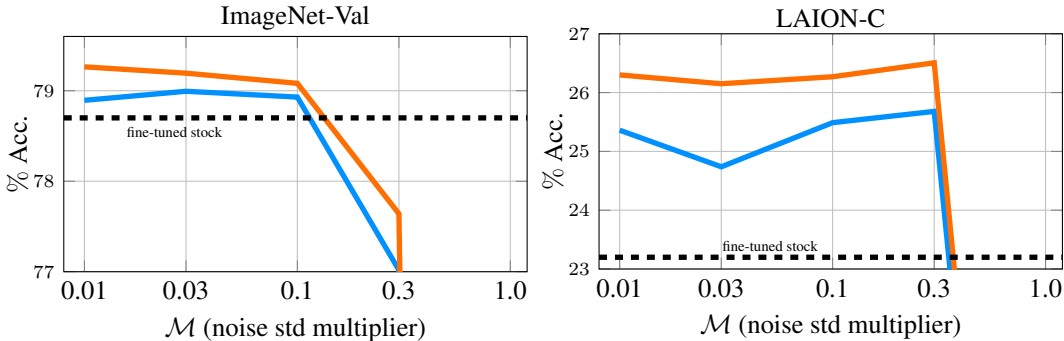

Figure 6: **Noisy soups are possible**. We directly fine-tune two models for each noise amount $\mathcal{M}$ added to the stock (blue), and we soup the two ingredients before testing. We also inter-train, then fine-tune two models for each noise amount added to the stock (orange), and also soup before testing. Robustness (right) peaks with greater noise. On both test sets, self-*soup*ervised inter-training outperforms direct fine-tuning. $\mathcal{M}=1.0$ adds too much noise, resulting in very low accuracy.

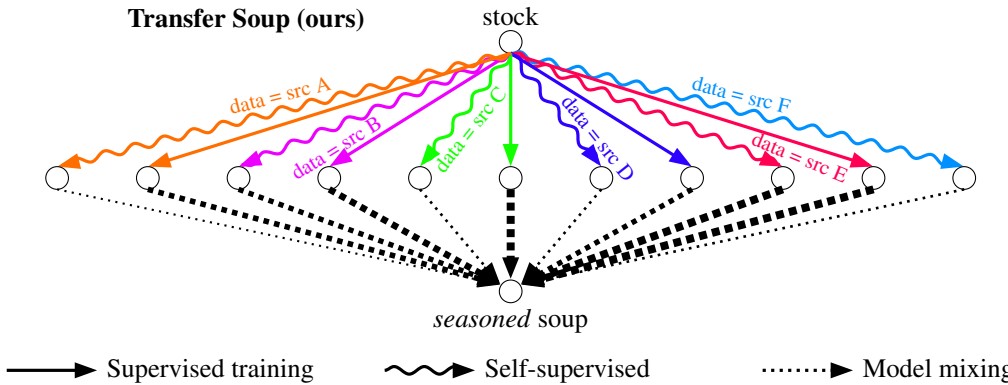

Figure 7: We introduce a recipe that scales the number of ingredients through more target tasks, with and without the use of labels. We show one use of these ingredients by "seasoning" them (Croce et al., 2023), which searches for mixture coefficients via sampling from the uniform distribution.

noise and then directly fine-tune on ImageNet. We perform this procedure twice and soup the two ingredients resulting from the same configuration (noise multiplier and training stages).

Figure 6 shows that noisy soups are possible: the stock need not be identical. For both self-souping and direct fine-tuning, IN-Val accuracy decreases at $\mathcal{M}=0.3$, and LAION-C accuracy *peaks* at $\mathcal{M}=0.3$, showing more parameter noise helps robustness.

## 4.5 SEASONING OVER SUPERVISION AND DOWNSTREAM TASKS

We select 17 datasets from the VTAB dataset set, which differ in terms of type (i.e., natural, specialized, and structured) and domain within a type (e.g., remote sensing and medical imaging are both specialized). For each dataset, we train two models initialized from the MAE stock; labels supervise one, and the other is self-supervised by MAE. To investigate whether representations learned on different tasks and losses can help a specific downstream task through souping, we "season"; i.e., we search for a convex combination of soup-mixture coefficients via randomly sampling from the uniform distribution. These coefficients weigh the corresponding ingredients' contribution to the soup, and we search over each downstream task. We run this protocol twice, once using all ingredients (supervised, self-supervised, and the stock), and again with SSL ingredients and the stock only.

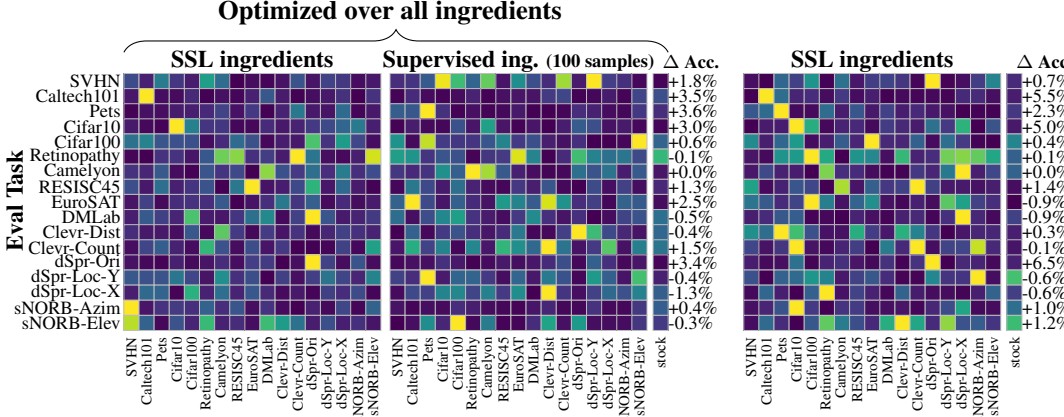

Figure 8: **Mixing ingredients from 17 tasks with and without supervision.** (Left) Mixtures optimized over supervised, SSL, and stock ingredients. (Right) Mixtures optimized over SSL and stock ingredients. Yellow = high, blue = low. Most tasks see gains: soups are possible and helpful when scaling across 17 different datasets and multiple losses. "Δ Acc." is the gain over the stock.

**Soups can use ingredients from different tasks *and* losses.** Figure 8 shows soups can be made from 35 diverse ingredients (all 17 datasets with and without supervision, plus the stock). Searching for the mixtures using 1K training points improves accuracy up to 6.5% over the stock.

## 5 RELATED WORK: SOUPS AND INTER-TRAINING

**Soups.** Our exotic model soups are informed by and add diversity to existing soups. Model soups (Wortsman et al., 2022) and DiWa (Ramé et al., 2022) concurrently discovered the possibility of souping. While equivalent in method, they differ in their experiments, and so are both informative and complementary. Model Ratatouille (Ramé et al., 2023) adds a stage of optimization, by inter-training the stock before fine-tuning, to increase the diversity of ingredients for domain generalization. We make use of inter-training, like Ratatouille, but while Ratatouille requires supervised inter-training on labeled data, we enable self-supervised inter-training on unlabeled data (including even the target data for inference). Furthermore, we experiment with inter-training supervised and self-supervised ingredients on a broader set of evaluations: the ImageNet variants and VTAB transfer benchmark. Model Stock (Jang et al., 2024) improves the fine-tuning efficiency of soups, in requiring fewer ingredients, by using the stock to inform layer-wise mixing. Our novel recipes are complementary to this improvement and the application of soups to different settings. Test-Time Ensemble (Kim et al., 2025) creates soups during test-time adaptation for improved robustness to new shifts. Soup to Go (Kleiman et al., 2025) applies soups to continual learning, which, unlike prior methods, does not require storing past data to prevent forgetting. Test-time and continual soups may be able to make use of our ingredients from self-soupervision.

**Inter-training without Mixing.** STILTs (Phang et al., 2018) inter-trains a pre-trained model on auxiliary tasks prior to fine-tuning on the target task. Hierarchical Pre-training (HPT (Reed et al., 2021)) inter-trains a pre-trained model via SSL prior to fine-tuning on the target task. In a sense, Model Ratatouille is to STILTs what our self-*soup*ervision is to HPT.

## 6 CONCLUSION

While model soups have been applied in more tasks and settings, with more ingredients and applications, the limits of linear mode connectivity and what can be souped or not have remained unclear. We show that a surprising diversity of soups are possible and productive: ingredients from different supervised and self-supervised losses, data partitions, and noisy parameters can be mixed to achieve equally good or better predictions on variety of image classification benchmarks. Our new ingredients and recipes can combine with existing soups, and so only multiply the number of soups.

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

# A APPENDIX

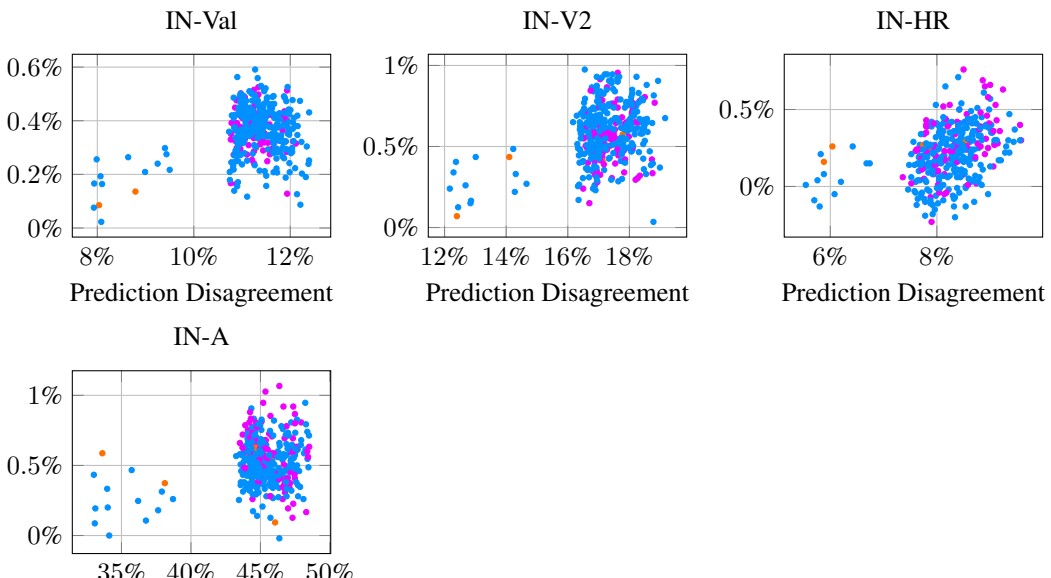

The plots above show two-ingredient soups over all combinations of test-set self-souping runs. Both ingredients were inter-trained on test data for orange points, one ingredient for pink points, and neither ingredient saw test data for blue points.

Table 3: Self-soupervision on unlabeled test data. "avg. ingredient" is computed by measuring the accuracy of each ingredient individually, then averaging the metrics.

| | Method | # ing. | IN-Val | IN-ReaL | IN-V2 | IN-HR | IN-A | IN-R | IN-C | LAION-C |
|---|---|---|---|---|---|---|---|---|---|---|
| **Vanilla Soups** | Avg. ingredient (LR) | 1 | 78.5 | 84.6 | 67.0 | 86.5 | 5.9 | 27.8 | 29.2 | 23.6 |
| | Soup (LR) | 3 | 79.2 | 85.4 | 67.6 | 87.2 | 6.2 | 29.5 | 31.4 | 27.6 |
| | Avg. ingredient (LR×WD) | 1 | 78.6 | 84.7 | 67.0 | 86.8 | 5.9 | 28.0 | 29.3 | 23.1 |
| | Soup (LR×WD) | 6 | 79.0 | 85.4 | 67.4 | 87.0 | 6.7 | 29.9 | 32.1 | 28.1 |
| | Avg. ingredient (LR×WD×SD) | 1 | 78.1 | 84.4 | 66.5 | 86.3 | 5.8 | 27.9 | 30.2 | 24.6 |
| | Soup (LR×WD×SD) | 12 | 78.0 | 84.8 | 66.8 | 86.3 | 6.5 | 29.2 | 33.9 | 30.0 |
| **Ingredients Inter-trained via SSL** | IN-Val avg. ingredient | 1 | 78.8 | 84.9 | 67.1 | 86.7 | 5.9 | 27.9 | 29.6 | 23.8 |
| | IN-Val soup | 4 | 79.1 +0.3 | 85.3 +0.4 | 67.7 +0.6 | 87.0 +0.3 | 6.3 +0.4 | 28.7 +0.8 | 31.7 +2.1 | 26.6 +2.8 |
| | IN-V2 avg. ingredient | 1 | 78.6 | 84.7 | 67.2 | 86.9 | 5.8 | 27.9 | 29.4 | 23.4 |
| | IN-V2 soup | 4 | 78.9 +0.3 | 85.2 +0.5 | 67.7 +0.5 | 87.0 +0.1 | 6.3 +0.5 | 28.8 +0.9 | 31.6 +2.2 | 26.5 +3.1 |
| | IN-HR avg. ingredient | 1 | 78.7 | 84.8 | 67.0 | 87.1 | 5.9 | 27.9 | 29.4 | 23.1 |
| | IN-HR soup | 4 | 79.0 +0.3 | 85.3 +0.5 | 67.5 +0.5 | 87.3 +0.2 | 6.2 +0.3 | 28.8 +0.9 | 31.7 +2.3 | 26.6 +3.5 |
| | IN-A avg. ingredient | 1 | 78.7 | 84.7 | 67.1 | 86.9 | 6.1 | 28.0 | 29.5 | 23.7 |
| | IN-A soup | 4 | 79.1 +0.4 | 85.3 +0.6 | 67.5 +0.4 | 87.1 +0.2 | 6.7 +0.6 | 28.9 +0.9 | 31.7 +2.2 | 26.6 +2.9 |
| | IN-R avg. ingredient | 1 | 78.7 | 84.7 | 67.3 | 87.1 | 5.9 | 29.5 | 29.7 | 23.0 |
| | IN-R soup | 4 | 79.2 +0.5 | 85.4 +0.7 | 67.9 +0.6 | 87.0 -0.1 | 6.3 +0.4 | 30.8 +1.3 | 31.8 +2.1 | 26.2 +3.2 |
| | IN-C avg. ingredient | 1 | 78.8 | 84.8 | 67.0 | 87.1 | 6.2 | 28.2 | 32.8 | 24.1 |
| | IN-C soup | 4 | 79.2 +0.4 | 85.3 +0.5 | 67.8 +0.8 | 87.4 +0.3 | 6.8 +0.6 | 28.9 +0.7 | 35.2 +2.4 | 27.3 +3.2 |
| | LAION-C avg. ingredient | 1 | 78.8 | 84.8 | 67.1 | 87.2 | 5.9 | 27.8 | 30.0 | 30.9 |
| | LAION-C soup | 4 | 79.1 +0.3 | 85.2 +0.4 | 67.8 +0.7 | 87.3 +0.1 | 6.3 +0.4 | 28.9 +1.1 | 32.1 +2.1 | 35.8 +4.9 |
| | Inter-trained avg. ingredient | 1 | 78.7 | 84.8 | 67.1 | 87.0 | 6.0 | 28.2 | 30.0 | 24.6 |
| | Inter-trained soup | 28 | 79.1 +0.4 | 85.3 +0.5 | 67.9 +0.8 | 87.0 +0.0 | 6.7 +0.7 | 29.6 +1.4 | 33.2 +3.2 | 28.9 +4.3 |
| | Inter-trained + vanilla avg. ing. | 1 | 78.5 | 84.7 | 66.9 | 86.8 | 5.9 | 28.1 | 30.1 | 24.6 |
| | Inter-trained + vanilla soup | 40 | 78.9 +0.4 | 85.2 +0.5 | 67.5 +0.6 | 86.9 +0.1 | 6.7 +0.8 | 29.5 +1.4 | 33.8 +3.7 | 29.5 +4.9 |

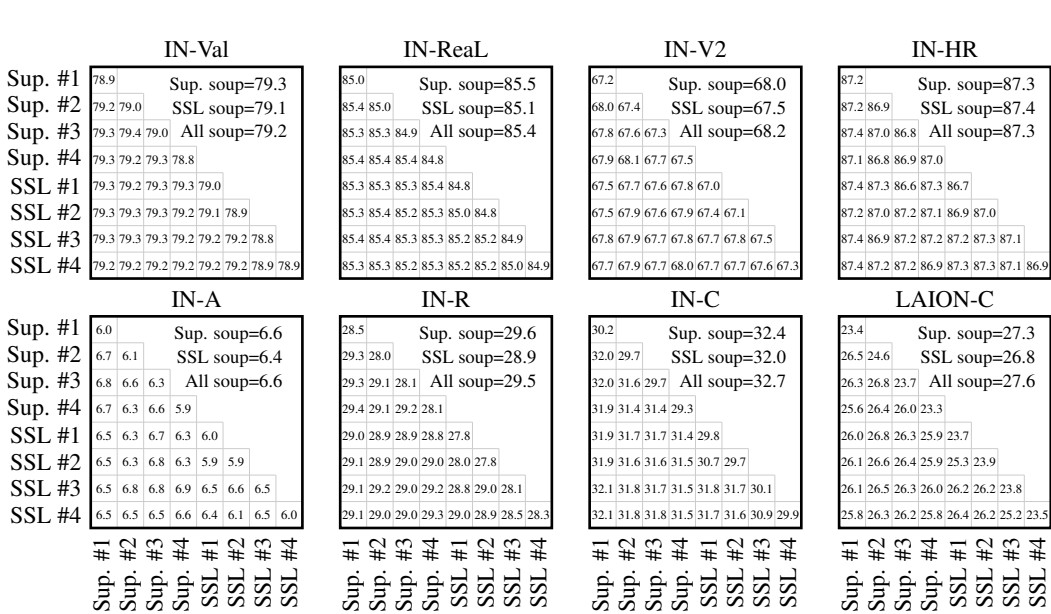

Figure 9: All two-ingredient partition soups tested on all the ImageNet test sets we consider.