# OpenReview forum: "Soup Kitchen: Mixing Exotic Model Soups across Labels, Losses, and Data"
_ICLR.cc/2026/Conference — Submitted to ICLR 2026_

### Official Review · Reviewer_ngdN · 2025-10-30

**Soundness:** 2
**Presentation:** 3
**Contribution:** 2
**Rating:** 4
**Confidence:** 4

**Summary:**

This paper explores extensions to the model souping framework for visual recognition. The authors investigate new ingredient types, including self-supervised ingredients (varying hyperparameters), soups mixed with supervised and self-supervised losses, soups mixed across tasks and data partitions, and ingredients fine-tuned on test data. Empirical results show improvements on ImageNet variants and VTAB.

**Strengths:**

The paper clearly outlines the proposed techniques and their benefits, making it easy to understand the core concept.

**Weaknesses:**

**(a) Limited Novelty:** While the paper extends model mixing to self-supervised training, the novelty is somewhat limited because it doesn't introduce new mixing strategies. The core mixing approach remains the same regardless of whether the ingredients are trained with or without labels.

**(b) Limited Scope of Experiments:** The empirical evaluation, while demonstrating improvements on ImageNet and VTAB, would be strengthened by including a broader range of datasets and tasks to better assess the generalizability of the proposed "exotic soups".

**(c) Test-Time Adaptation Concerns:** The self-supervision on test data, while yielding performance gains, raises significant concerns regarding potential overfitting or data leakage. The contribution of this specific technique is questionable and requires further justification and analysis to ensure it doesn't compromise generalization.

**Questions:**

See Weaknesses.

---

### Official Review · Reviewer_9Ynr · 2025-10-31

**Soundness:** 3
**Presentation:** 3
**Contribution:** 2
**Rating:** 6
**Confidence:** 3

**Summary:**

The paper studies model soup, the pipeline in which a model (the stock) is finetuned in different ways to different models which are then mixed together into a soup model with improved performance. It extends previous works by considering new ways to finetuning and mixing with self-supervision. These include SSL training on different test sets then fine-tuning on target training set, training on different parts of the target training set in a self-sueprvised or supervised manner before finetuning on the whole set, and adding noise into the model at various points during the pipeline.

The paper presents extensive experiments on these different options and show that they all are effective and bring performance improvements.

**Strengths:**

The paper's study on the effectiveness of using self-supervised training in model soup in interesting. The proposed new ways to fine-tune the stock are sound. The proposal to add noises into the model before finetuning is also reasonable.

This is a heavily experimental paper. It provides extensive experiments on different finetuning /mixing techniques to prove their effectiveness. The design of the experiments are sound and for each proposed technique, enough empirical evidence are provided to support it. Though there is no theoretical support provided for model souping in the paper, Figure 3 showing evidences of the linear mode connectivity is interesting and provides some assurance.

The paper is well written in general.

**Weaknesses:**

Since there is no theoretical ground for model souping provided, I am not sure insights obtained from the experiments in the paper hold in other settings. It confirms that it is possible to soup in certain settings but to know for sure if model souping still works in a specific cases.

Comparisons provided in the paper are often between the proposed soup and vanilla soup. I wonder how the proposed soup fares against a very well tuned model. Even comparing to vanilla soup, the gain of the proposed soup is not universal: Gains are observed on some benchmarks but not all (Tables 1 and 2). Also, certain gains observed are not clearly due to souping. For example, gains on LAION-C in Table 1 apparently comes from finetuning on LAION-C. I am not sure how big the role of souping is in this case.

**Questions:**

I would like to see a comparison between the proposed soup and a very well tuned model. Showing that proposed soups are better than a well tuned model will strengthen the paper.

For soups where the stock is first finetuned on different test sets, I think it is important to show the model's performance before mixing to disentabgle the effect of souping and test set fine-tuning.

---

### Official Review · Reviewer_fEEU · 2025-11-01

**Soundness:** 2
**Presentation:** 2
**Contribution:** 2
**Rating:** 4
**Confidence:** 3

**Summary:**

This paper introduces improved strategies for model soups. It first fine-tunes a pre-trained model into several middle checkpoints, and then mixes them to form a single model to improve predictions. The main innovation of this paper is to apply self-supervised training in the first fine-tuning stage. It also suggests adding noise to obtain more diverse models. For model mixing, the paper proposes to mix representations instead of the predictions. The paper conducts extensive experiments on ImageNet variants and the VTAB benchmark to evaluate the proposed self-soupervision framework. The results show that model soups are partially effective by using unlabeled, partitioned, or noisy data. The experiments also confirm that linear mode connectivity can hold across self-supervised models and heterogeneous tasks.

**Strengths:**

1. Model soup is an interesting method which improves model generalization without extra inference cost.
2. The paper investigated several aspects to improve model soups, including self-supervised training, noisy soups and representation mixing, which are complementary to previous work.

**Weaknesses:**

1. The main contributions are a bit incremental, as these advanced strategies are based on the existing pipeline of model soups. The self-supervised learning (SSL) part, which is this paper’s main technical contribution, looks straightforward since SSL is well-known to have better generalization capacity than supervised learning.
2. The idea of employing self-souping with unlabeled test data is problematic. In many real-world applications, test data are totally unseen data (e.g., further user queries in a search engine) and therefore it’s not realistic to assume the availability of unlabeled test data. The authors explain that DA and TTA also optimize models using test data. However, these are special settings which are different from standard machine learning. If the research focuses on the scenarios that test data are known but have domain gaps with training data, it’s necessary to provide comprehensive literature review from DA or TTA papers to discuss optimal methods of employing unlabeled test data.
3. The motivation and rationale of representation mixing are not clear. Please explain why this way is better than prediction mixing. The paper mentions that labels are not needed in representation mixing. However, the labels are available since the target tasks are supervised learning tasks. Why is this label-free nature an advantage?
4. The symbols $x_i, y_i$ are not proper to define datasets, as they are usually used as instances. Using $X_i, Y_i$ is more common.
5. Table 2 is not referred to in the main paper. It seems to correspond to Section 4.3. But the results are not explained, and the overall improvements are not significant.
6. The experiments mainly show the effectiveness of self-soup (especially when using in-domain test data), but the state-of-the-art baselines are not evaluated and compared against the proposed method (only vanilla soup is evaluated).

**Questions:**

1. In Figure 3, what do the orange and blue colors mean?
2. In line 310, how do you get the improvement numbers, i.e. 3.4%, 7.7%, 12.6%?
3. Which experiment represents the results by using all the proposed components (SSL+noisy+seasoning)?

---

### Official Review · Reviewer_xzSe · 2025-11-01

**Soundness:** 2
**Presentation:** 2
**Contribution:** 1
**Rating:** 2
**Confidence:** 4

**Summary:**

This paper extends model soups (weight averaging of multiple fine-tuned models) beyond the usual supervised setting. It introduces self-soupervision: creating “ingredients” via self-supervised learning (SSL), then mixing them—sometimes together with supervised ingredients—to form a single “soup” that improves accuracy and robustness without extra inference cost. Key claims: (1) ingredients can be produced without labels (varying SSL hyperparameters); (2) soups can mix across losses (e.g., MAE with MoCoV3); (3) soups can mix across 10+ data distributions and target-set partitions; and (4) ingredients can be SSL-adapted on the (unlabeled) test data before soup mixing. Reported gains are ~1–3% robustness on ImageNet variants and up to 10% transfer on VTAB.

**Strengths:**

*

**Weaknesses:**

* Major concerns
    * Limited novelty
        * While expanding the scope of model merging to self-supervised models are meaningful. However, this paper seems to naively expand previous model merging research to additional self-supervised models without important intuition. Specifically:
            * As far as I understand, the core motivation of Soup Kitchen is that previous studies limit the scope of model merging only to supervised model, thus expand the model scope to self-supervised models.
            * If the authors claim an important challenges about why the previous studies did not employ the self-supervised models for their methods, and if the authors suggest reasonable solutions to the challenges, then the contributions of this paper would be impactful and novel. However, the method section mainly emphasize the alleviation of the dependency on labeled data
        * Generalizability of the proposed Soup Kitchen to self-supervised models seems not to be sufficiently proven
            * The number of employed self-supervised models is too small to claim that this paper has successfully expanded model merging approaches to self-supervised models. Proving the proposed method only on two pre-trained models (e.g., MAE and MoCo v2) is insufficient
            * I recommend to further prove the generalizability of the proposed method. There exist a lot of self-supervised models and I also can suggest at least 20 self-supervised models pre-trained on vision domain. Nevertheless, I'll simply refer some important works in the self-supervised learning area [1-9].
    * Comparison with previous model merging methods are missing
        * Though the proposed method focus on merging self-supervised models, Soup Kitchen should be compared to previous model merging methods to validate its effectiveness. Moreover, since the authors have tried to merge both self-supervised models and supervised models together. This makes more essential to provide comparisons to prior work,


[1] Grill et al., BYOL, NeurIPS 2020

[2] Caron et al., DINO, ICCV 2021

[3] Bao et al.,  BEIT, arXiv preprint arXiv:2106.08254, 2021

[4] Peng et al., BEIT v2, arXiv preprint arXiv:2208.06366

[5] Xie et al., SimMIM, CVPR 2022

[6] Assran et al., I-JEPA, ICCV 2023

[7] zhou et al, iBOT, ICLR 2022

[8] Baevski et al., data2vec, ICML 2022

[9] Baevski et al., data2vec2.0, ICML 2023

**Questions:**

.

---

### Meta-Review · Area_Chair_p7Fq · 2026-01-07

**Summary:**

The reviewers' opinion leans toward rejection, and the authors don't submit rebuttal comments. Therefore, I can't recommend acceptance for the paper.

**Reviewer Concerns:**

The rebuttal is not submitted.

**Reviewer Scores:**

The rebuttal is not submitted.

---

### Decision · Program_Chairs · 2026-01-26

Reject